# *Hericium erinaceus* Extract Exerts Beneficial Effects on Gut–Neuroinflammaging–Cognitive Axis in Elderly Mice

**DOI:** 10.3390/biology13010018

**Published:** 2023-12-28

**Authors:** Erica Cecilia Priori, Daniela Ratto, Fabrizio De Luca, Anna Sandionigi, Elena Savino, Francesca Giammello, Marcello Romeo, Federico Brandalise, Elisa Roda, Paola Rossi

**Affiliations:** 1Department of Biology and Biotechnology “L. Spallanzani”, University of Pavia, 27100 Pavia, Italy; ericacecilia.priori@unipv.it (E.C.P.); daniela.ratto@unipv.it (D.R.); fabrizio.deluca@unipv.it (F.D.L.); francesca.giammello01@universitadipavia.it (F.G.); drmarcelloromeo@gmail.com (M.R.); 2Department of Biotechnology and Biosciences, University of Milano-Bicocca, 20126 Milan, Italy; anna.sandionigi@quantiaconsulting.com; 3Quantia Consulting S.r.l., Via Petrarca 20, 22066 Mariano Comense, Italy; 4Department of Earth and Environmental Science, University of Pavia, 27100 Pavia, Italy; elena.savino@unipv.it; 5Department of Biosciences, University of Milan, 20133 Milano, Italy; federico.brandalise@unimi.it; 6Laboratory of Clinical & Experimental Toxicology, Pavia Poison Centre, National Toxicology Information Centre, Toxicology Unit, Istituti Clinici Scientifici Maugeri IRCCS, 27100 Pavia, Italy

**Keywords:** *Hericium erinaceus*, gut microbiota, gut–brain axis, neuroinflammaging, frailty, ageing, hippocampus

## Abstract

**Simple Summary:**

The gut microbiome is a complex and unendingly changing community of bacteria that lives inside animals, including humans. Emerging evidence has proved that gut microbiome composition is associated with several human health outcomes, which include cognitive performance. However, few epidemiological studies exist, and research in this field is still ongoing and developing. The current article suggests that oral consumption of an edible medicinal mushroom named *H. erinaceus* promotes the growth of beneficial gut bacteria, parallelly reducing pathogen bacteria, therefore revealing its prebiotic effect. Additionally, this oral supplementation had a positive impact on cognitive function, also leading to a decrease in inflammation in the hippocampus, a brain area crucially involved in memory formation and consolidation. Overall, these findings support the notion that changing the gut microbiome composition through nutrition modulation could trigger longevity-promoting effects, protecting from age-related cognitive decline.

**Abstract:**

Ageing is a biological phenomenon that determines the impairment of cognitive performances, in particular, affecting memory. Inflammation and cellular senescence are known to be involved in the pathogenesis of cognitive decline. The gut microbiota–brain axis could exert a critical role in influencing brain homeostasis during ageing, modulating neuroinflammation, and possibly leading to inflammaging. Due to their anti-ageing properties, medicinal mushrooms can be utilised as a resource for developing pharmaceuticals and functional foods. Specifically, *Hericium erinaceus* (He), thanks to its bioactive metabolites, exerts numerous healthy beneficial effects, such as reinforcing the immune system, counteracting ageing, and improving cognitive performance. Our previous works demonstrated the capabilities of two months of He1 standardised extract oral supplementation in preventing cognitive decline in elderly frail mice. Herein, we showed that this treatment did not change the overall gut microbiome composition but significantly modified the relative abundance of genera specifically involved in cognition and inflammation. Parallelly, a significant decrease in crucial markers of inflammation and cellular senescence, i.e., CD45, GFAP, IL6, p62, and γH2AX, was demonstrated in the dentate gyrus and *Cornus Ammonis* hippocampal areas through immunohistochemical experiments. In summary, we suggested beneficial and anti-inflammatory properties of He1 in mouse hippocampus through the gut microbiome–brain axis modulation.

## 1. Introduction

Ageing is characterised by the decline of physiological functions, affecting both animals and humans. Given the increasing global population of elderly individuals, the promotion of healthy ageing has become an area of critical focus [1,2]. Recent studies are focused on developing therapeutic methods that seek to identify natural substances that can potentially counteract or reverse the adverse effects of ageing, including frailty [3]. Frailty is a multifaceted condition that affects the elderly population, characterised by a decline in bodily functions and physiological reserves across multiple systems, elevating the risk of negative health outcomes [4]. In particular, cognitive frailty refers to a condition in which an individual exhibits physical dysfunction associated with mild cognitive deficits [5] due to brain ageing. Brain ageing can lead to cognitive impairment, motor disorders, and emotional disturbances caused by various morphological and functional changes in the brain [6]. In particular, episodic and recognition memories are known to decline during brain ageing. Recognition memory is essential for animals, including humans, and is considered a key aspect of personality [7] and is dependent on the hippocampal area. The proper functioning and structure of the hippocampus are essential for normal learning and memory consolidation, and this region is especially susceptible during ageing [8]. Literature data have shown that changes in the structure and function of the hippocampus are linked to the severity and progression of cognitive decline-related neurodegenerative disorders in both humans and animals [9], and it is known that, during ageing, the hippocampus undergoes various morphological and cellular-level changes [10,11,12]. Furthermore, during ageing, there is a decrease in hippocampal neurogenesis that can lead to a decrease in synaptic plasticity, which heavily depends on the generation of new neurons [13,14]. Additionally, both experimental and epidemiological evidence have indicated that the ageing brain, including the hippocampus, shows an increase in reactive gliosis and gliogenesis [15,16,17]. In fact, the contribution of glial cells to injuries and degenerative conditions in the central nervous system (CNS) is widely recognised, and it is confirmed by the presence of activated astrocytes and microglial cells and the release of soluble cytokines.

Hence, the inflammatory processes could have a vital impact on the complex pathophysiological interactions observed in the ageing process [18]. 

There is growing evidence linking frailty to immunosenescence and chronic systemic inflammation, also known as inflammaging, a hallmark of accelerated ageing [19,20]. Furthermore, inflammaging increases the risk of several conditions, like cardiovascular diseases, cancer, dementia, cognitive decline, physical disabilities, and frailty [19,20]. Inflammaging and cellular senescence have been found to play a crucial role in the development of cognitive impairments and age-related neurological disorders, as demonstrated by cellular, biochemical, and molecular studies [21,22,23]. Inflammaging and, in general, increased inflammation can arise from gut microbiota dysbiosis. Indeed, inflammation and gut microbiota are interconnected in a complex relationship, where the imbalance of the latter can trigger a cascade of inflammatory responses within the body [24]. Recent research demonstrated that the gut microbiota composition undergoes significant changes during ageing, which strongly correlates with age-related ailments like frailty [25,26,27]. Indeed, both preclinical and clinical studies demonstrated that elderly individuals display a different gut microbiome composition compared with their younger counterparts [28,29].

The connection between the gut microbiota and the brain is closely intertwined through a two-way communication system referred to as the brain–gut–microbiota axis [30]. Disruption of this axis is a crucial factor in the cognitive decline associated with ageing; it is essential to maintain a healthy gut–microbiota–brain axis so that normal cognitive functioning is preserved over time [31,32]. Indeed, the importance of maintaining eubiosis of gut microbiota to safeguard cognitive performances in the population is an actual and critical point, also during physiological or accelerated ageing [33]. 

*Hericium erinaceus*, commonly referred to as Lion’s mane and Monkey Head Mushroom, is a type of mushroom that is both edible and medicinal [34]. It is highly regarded for its exceptional health-promoting properties, and it can typically be found in northern temperate latitudes, including Italy [35]. It produces more than 70 bioactive metabolites, including β-glucans, erinacines, and hericenones, conferring to *H. erinaceus* various health-promoting properties, such as anticancer, antioxidant, anti-senescence, neuroprotective, and antidepressive activities [36,37]. For all these beneficial properties, in this paper, we investigated its potential as a therapeutic method for counteracting or reversing the adverse effects of ageing, including frailty. In particular, we explored the potential of *H. erinaceus* in modulating the gut microbiota–brain axis in mice during ageing, focusing on the interaction among *H. erinaceus* oral supplementation, cognitive frailty, gut microbiome composition, CNS inflammaging, gliosis, and cellular senescence. 

## 2. Materials and Methods

### 2.1. Animals 

Before starting the experimental procedures, fifteen C57BL-6J male mice free from pathogens were acclimated for approximately 30 days at the Animal Care Facility of the University of Pavia. These animals were obtained from Charles River, Italy. Throughout the duration of the experiment, the mice were housed in an environment with a constant temperature of 21 ± 2 °C and humidity of 50 ± 10%. Additionally, they were subjected to a controlled 12 h cycle of light and darkness, and they had ad libitum access to water and food. All the experiments were carried out in accordance with the guidelines laid out by the institution’s animal welfare committee, the Ethics Committee of Pavia University (Ministry of Health, Licence number 774/2016-PR), also in compliance with the European Council Directive 2010/63/EU.

### 2.2. Experimental Plan, Behavioural Tests, and Cognitive Frailty Index Measurement 

The current study is an essential part of a previous broader study aimed at analysing the changes during the murine life span, from 11 (T0, adulthood) to 23.5 months old [29,38]. In the current study, we placed the focus on 21.5 (T1, early senescence) and 23.5 (T2, late senescence) months. The alpha and beta diversity data and gut microbiota composition before He1 treatment were taken from our previous investigation [29]. Animal faecal stool samples were collected and stored at −80 °C. Parallelly, the estimation of mouse cognitive performances were achieved [38,39]. Indeed, at each experimental session, mice performed two spontaneous behavioural tests, i.e., Emergence and Novel Object Recognition (NOR) tasks, to evaluate their recognition memory performances through SMART video tracking system (2 Biological Instruments, Besozzo, Varese, Italy) and Sony CCD colour video camera (PAL). For all experiments, the researchers were unaware of which group (control and He1) the mice belonged to. Briefly, we performed emergence and NOR tests as reported in Brandalise et al., 2017 [40] and Ratto et al., 2019 [38]. In particular, for evaluating recognition memory, we focused on specific parameters in the two different tests: the number of exits, the latency of first exit, the time of exploration outside in the emergence test, and the Mean Novelty Discrimination Indices of the number and of the time of approaches in the NOR test. For each parameter, we calculated the respective Cognitive Frailty Index (FI) using the formula and the values reported in Ratto et al., 2019 [38]: FI = (Value − Mean Value at T0)/(Standard Deviation at T0) × ±0.25

Then, we averaged each single FI value of each specific parameter to obtain a specific cognitive FI for emergence and for the NOR test. Then, averaging the two Cognitive FI scores, we obtained a global Cognitive FI that allowed us to evaluate the recognition memory performances. Notably, the global Cognitive FI scores of each specific mouse were used for selecting the mice as healthy or frail animals. Indeed, FI values at T1 allowed us to identify the seven frailest mice that constituted the He1 group, which, for a period of two months from T1, were supplemented with a drink composed of *H. erinaceus* blend containing standardised sporophore and mycelium ethanol extracts (He1) solubilised in water (1 mg dry weight of supplement/mouse daily). The chosen amount was intended to simulate the oral supplementation doses commonly used by humans [38]. The remaining mice constituted the control (not-supplemented) group. 

All mice were euthanised at T2 when brain tissue sampling was performed for further histochemical, immunohistochemical, and immunofluorescence evaluations.

### 2.3. Hericium erinaceus Extracts: Content and Metabolites

As previously reported, strain 1 of *H. erinaceus* (He1, actually stored in MicUNIPV) was obtained from a wild sporophore found in Siena province, Tuscany, Italy in 2013 [38]. The extraction procedures were described in Cesaroni et al. 2019 [35], Corana et al. 2019 [41], and Ratto et al. 2019 [38]. The He1 metabolites were identified and measured through HPLC-UV-ESI/MS analyses, previously described, using specific standards and accurate calibration curves.

### 2.4. Bacterial DNA Extraction, 16s rRNA Sequencing, Illumina Data Processing, and Gut Microbiome Characterisation

The bacterial DNA extraction and 16s rRNA gene sequencing were performed, as reported in Ratto et al., 2022 [29]. Briefly, the mouse stools were extracted and quantified. Next, we used the specific forward (V3 F: 5′-TCGTCGGCAGCGTCAGATGTGTATAAGAGACAG-3’) and reverse (V4 R: 5’-GTCTCGTGGGCTCGGAGATGTGTATAAGAGACAG-3’) primers in PCR analysis to prepare the amplicons for the sequencing by MiSeq Illumina, carried out by the BMR Genomics SRL of Padova. 

### 2.5. Necropsy and Brain Specimen Preparation

At T2, 23.5-month-old mice were sacrificed, as previously reported [17,38]. After anaesthetisation, brains were immediately excised, fixed for 48 h in 4% paraformaldehyde in 0.1 M phosphate buffer (pH 7.4), dehydrated in ethanol followed by acetone, and embedded in Paraplast X-TRA (Sigma Aldrich, Milan, Italy). Eight micrometre-thick brain sections were cut in the coronal plane and collected on silane-coated slides. 

### 2.6. Haematoxylin and Eosin (H&E) Staining

The evaluations focused on the hippocampus. To analyse the gross morphology and neuronal cytoarchitecture of dentate gyrus (DG) and Cornus Ammonis (CA) hippocampal areas by light microscopy, H&E staining was performed as previously reported [39,42]. Briefly, approximately 20 randomised sections (5 microscopic fields) per animal and time/condition were examined by a blinded operator using a Leica DM6B WF microscope (Leica Microsystems, Buccinasco, MI, Italy). The images were acquired with Leica dfc 7000 t CCD camera (Leica microsystems, Buccinasco, MI, Italy) and stored on a PC running the Leica Application Suite X (LAS X) software (Version 5.1.0).

### 2.7. Picrosirius Red (PSR) Staining

Coronal sections of the hippocampus were subjected to staining using a solution of PSR (0.1% Sirius Red dissolved in saturated aqueous picric acid) for a duration of 1 h. Following this, the sections were rinsed in 5% acidic water for staining of collagen bundles [43]. Next, sections were dehydrated in ethanol, cleared in xylene, and finally mounted using Eukitt (Kindler, Freiburg, Germany).

### 2.8. Immunohistochemical and Immunofluorescence Assessment and Quantitative Evaluations

The immunohistochemical and immunofluorescence assessment and subsequent quantitative evaluations were performed as previously published [17,38]. Concisely, commercial antibodies were used to perform immunohistochemical experiments on murine hippocampal samples to explore the expression and distribution of selected markers: Cluster of Differentiation 45 (CD45), p62, Glial Fibrillary Acidic Protein (GFAP), Interleukin 6 (IL6), and H2A histone family member X (γH2AX). Hippocampal sections were placed in a dark, moist chamber at environmental temperature and incubated overnight with PBS-diluted primary antibodies (listed in Appendix A).

Brightfield microscopy—immunohistochemistry: appropriate biotinylated secondary antibodies (listed in Appendix A) and an avidin biotinylated horseradish peroxidase complex from Vector Laboratories (Burlingame, CA, USA) were utilised to identify antigen/antibody interaction site. 3,3′-diaminobenzidine tetrahydrochloride peroxidase substrate was employed as the chromogen (Sigma-Aldrich, St. Louis, MO, USA). Carazzi’s Haematoxylin allowed nuclear counterstaining. Then, sections were dehydrated using ethanol, cleared with xylene, and finally mounted in Eukitt (Kindler, Freiburg, Germany). Next, sections were observed with a Leica DM6B WF microscope (Leica microsystems, Buccinasco, MI, Italy), and images were acquired with a Leica dfc 7000 t CCD camera (Leica microsystems, Buccinasco, MI, Italy).

Immunofluorescence: sections were incubated with primary antibodies (1 h, environmental temperature) (Appendix A) and, subsequently, with secondary antibodies (Appendix A). Then, nuclei were stained with 0.1 μg/mL Hoechst 33258 (Sigma Aldrich, Milan, Italy). After washing, coverslips were added with Mowiol (Calbiochem, San Diego, CA, USA). 

Slices were examined using a Leica DM6B WF microscope (Leica microsystems, Buccinasco, MI, Italy), images were captured with an ORCA-Flash4.0 V3 Digital CMOS camera C13440-20CU (Hamamatsu Photonics, Arese, MI, Italy), and results were analysed using the Leica Application Suite X (LAS X) software (Version 5.1.0).

To prevent potential discrepancies in results caused by slight procedural variations, all immunostaining reactions were performed simultaneously on slides from different experimental groups. As control, some sections were incubated without primary antibodies, using only PBS; immunoreactivity was observed under this condition.

The extent of histochemical, immunohistochemical/immunofluorescence labelling was evaluated on section images obtained from exposure times that avoid any pixel saturation. The labelling intensity, measured as optical density (OD), was determined through densitometric analysis, following previously reported methods [39]. Precisely, the OD was measured in 3 randomly chosen images/sections, with at least 10 measurements performed per image for 5 photographs/animal in each experimental group. Data were recorded using Excel software, and the analysis was conducted using Image-J 1.48i software (NIH, Bethesda, MA, USA).

The following measurements were conducted for each hippocampal area (DG and CA subfields): (i) density count of shrunken cells using a 40× objective on slides stained with H&E (number of cells/mm^2^); (ii) density count of CD45/p62/GFAP/IL-6/γH2AX-immunopositive cells (number of immunopositive cells/mm^2^); (iii) density count of GFAP/IL-6 colocalisation (number of double immunopositive cells/mm^2^).

### 2.9. Statistics

Data were presented as the mean ± standard error of the mean (SEM). We conducted Bartlett and Shapiro–Wilk Tests to verify the normal distribution of the parameters. To evaluate significant differences in cognitive frailty values, we employed unpaired Student’s *t*-test for comparing two groups or One-Way ANOVA for comparing three groups. To evaluate statistically significant differences in immunostaining experiments between control and He1-supplemented mice, we employed unpaired Student’s *t*-test. All these statistical analyses were conducted using GraphPad Prism 7.0 software (GraphPad Software Inc., La Jolla, CA, USA). Regarding gut microbiome study, the analysis was performed according to Ratto et al., 2022 [29]. 

Significance was considered at *p* < 0.05 (*), *p* < 0.01 (**), *p* < 0.001 (***), *p* < 0.0001 (****). 

## 3. Results

### 3.1. Metabolites in Hericium erinaceus Extract (He1)

By means of HPLC-UV-ESI/MS and by using standards, we identified and quantified the amount of different metabolites in 70% ethanol *Hericium erinaceus* extracts of mycelium and sporophore (strain He1) (see Methods, Section 2.3). Specifically, the He1 sporophore extract contained 500 µg/g Hericenone C, less than 20 µg/g Hericenone D, and 340 µg/g L-Ergothioneine while the mycelium extract contained 150 µg/g Erinacine A and 580 µg/g L-Ergothioneine (Table 1).

### 3.2. Cognitive Frailty Index as Selection Criterion for Mice Recruitment 

Behavioural tests in vivo were performed in early and late senescent mice (21.5 and 23.5 months old, T1 and T2). The recognition memory (knowledge component) was tested by using NOR and emergence tests, and the FI index scores were calculated as previously described (see Methods and Ratto et al. 2019 [38], Ratto et al. 2022 [29]). Individual FI scores at T1 allowed us to identify the seven frailest mice by setting a threshold FI value of 1.3. The frailest mice constituted the He1 group (*n* = 7) that we called at T1, *pre-He1* supplementation (T1 pre-He1) and at T2, post-He1 supplementation (T2 He1; Figure 1A) groups. The remaining mice constituted the control not-supplemented group (Figure 1A). Notably, the frailty index between control and *pre-He1* mice was statistically different at T1 (*p* = 0.0018; Figure 1A). The frailest mice were supplemented for sixty days (from T1 and ending to T2) with a drink composed of an *H. erinaceus* blend containing sporophore and mycelium ethanol extracts of known composition (Table 1) solubilised in water to supply 1 mg He1 per mouse daily. At T2, the behavioural tests were repeated, and the FI index score was calculated. Figure 1B shows the Cognitive frailty index of the knowledge component before and after He1 supplementation in untreated (T2 CNTR) and treated (T2 He1) mice. It should be noted that the bettering of the cognitive frailty index after He1 supplementation in the two experimental groups at T2 was not statistically significant, according to what was previously described in averaged data [38]. In particular, all of the frailest mice displayed a recovery of the knowledge component of the recognition memory after 2 months of oral supplementation with the He1 blend.

### 3.3. The Effect of He1 Treatment on the Gut Microbiome Composition during Ageing

We studied the effect of He1 standardised extract oral supplementation during ageing on the gut microbial communities using 16S ribosomal RNA (rRNA; hypervariable regions V3–V4) gene sequencing [29]. After quality filtering, merging reads, and chimaera removal, we obtained 775,708 sequences (median frequency = 37,580 reads per sample). We identified 1858 amplicon sequence variants (ASVs). The Appendix A represents the rarefaction curve.

Looking at the overall microbiota composition, the alpha-diversity (Faith phylogenetic metrics) at T1 was significantly lower in pre-treated frailest He1 mice compared to healthy control mice (*p* < 0.05), confirming that the gut microbiome reflects the mice’s cognitive performances [29]. Notably, after 60 days of He1 treatment, there was a non-significant trend in alpha-diversity increase (*p* = 0.9; Figure 2a). Regarding the β-diversity observed in non-metric multidimensional scaling (NMDS) analysis, there were no different clusters in the control and He1 groups at T1 and T2 (Figure 2b). Therefore, the two-month oral supplementation with He1 did not significantly change the overall gut microbiome composition. However, compared to control mice, the He1 treatment significantly reduced the relative abundance of *Odoribacter, Clostridia vadinBB60*, and *Muribaculaceae* and significantly increased the relative abundance of genera *Clostridia UCG-014, Lachnospiraceae_NK4A136*, and *Eubacterium xylanophilum* (Figure 2c). 

### 3.4. Light Microscopy Evaluation and Immunohistochemical Study

All microscopy experiments were conducted on coronal brain sections from aged control animals and He1-treated mice at T2 (23.5-month-old animals), concentrating on the hippocampus since this CNS area is critical in recognition memory [44,45]. All investigations focused on the DG and CA regions (including CA subdivisions), which are highly susceptible to neurodegeneration during ageing and inflammation [46].

#### 3.4.1. He1 Supplementation Preserves Healthy Hippocampus Cytoarchitecture

To assess the possible incidence of age- and/or He1-related alterations in the cytoarchitecture of the hippocampus in old mice, H&E staining was performed. The overall physiological general morphology of the hippocampus was conserved both in control and He1-treated mice (Figure 3). The CA region was typically divided into four areas—CA1, CA2, CA3, and CA4—while the DG region formed a V-shape, which included the CA4 area. Both controls and He1-treated mice exhibited the characteristic three layers in the CA, namely the polymorphic layer (POL), pyramidal cell layer (PYL), and molecular layer (ML). Likewise, the DG consisted of three well-defined strata: molecular layer (ML), granule layer (GL), and polymorphic layer (POL), with the latter representing the hilus. Concerning the DG, in both control and He1-treated animals, the GL contained densely packed granule cell bodies. Notably, a greater density of shrunken cells was assessed both in the DG, mainly localised in GL and PL, as well as in the CA region of control animals compared to He1-treated mice (Figure 3).

#### 3.4.2. Picrosirius Red Staining: Fibrillar Collagen Network Evaluation

The Picrosirius Red staining method was used to evaluate collagen fibre organisation in paraffin-embedded tissues [47]. Both in controls and He1-treated mice, collagen fibres were mainly detected in collagen-rich vascular wall ECM, constituting the inner cellular lining of hippocampal blood vessels, showing a marked PSR staining. Notably, both in the DG as well as in the CA subfields, the quantitative investigation evidenced a significantly higher collagen fibre optical density (OD) in control mice vs. He1-treated animals (Figure 4, *p* < 0.01 and *p* < 0.0001 for DG and CA, respectively). 

#### 3.4.3. He1 Supplementation Decreases Microglia Activation

CD45 is a cellular marker highly expressed in microglia upon inflammation and ageing [48]. Both in controls and He1-treated mice, CD45 immunopositivity was detected in the DG and in four CA subfields, i.e., CA4, CA3, CA2, and CA1. Notably, the heaviest CD45 immunoreactivity was observed almost exclusively in control animals. In regard to the DG region, strongly CD45-immunopositive cells were clearly distinguished, primarily localised in the thickness of the GL. Several markedly CD45-immunoreactive granular cells located near the SGZ and in the PL were evidenced in control mice only. Interestingly, the same marked immunoreactivity was statistically significantly higher in controls compared to He1-treated mice in all CA regions. Specifically, several CD45-immunopositive cells were detected in the CA4 and CA2 regions of control animals, while immunoreactivity was completely lacking in He1-treated mice. In both experimental groups, the CA3 region exhibited pale immunolabeling (Figure 5). Accordingly, the quantitative analysis, comparing He1-treated mice with controls, documented an extremely significant reduction of CD45-immunoreactivity in the DG, measured in terms of both immunopositive cell density (Figure 5, GL: *p* < 0.0001 and PL: *p* < 0.01) and OD (Figure 5, GL: *p* < 0.01 and PL: *p* < 0.0001). Similarly, regarding all the CA subregions, a significant lessening of CD45-immunolabeling, assessed in terms of both CD45-immunopositive cell density and OD, was recorded comparing He1-treated mice with control animals (Figure 5, *p* < 0.0001).

#### 3.4.4. He1 Supplement Reduces Neuroinflammaging

The distribution of specific molecules, i.e., GFAP and IL6, have been immunohistochemically assessed as typical markers of reactive gliosis and inflammatory pathways [23,49]. A spreading GFAP expression was detected both in the DG and CA area of controls and He1-treated mice. In detail, in control animals, a layer of GFAP-immunopositive astrocytes was observed, characterised by noticeably thickened and markedly immunoreactive soma and extensions, primarily located in the PL of DG. Contrarily, concerning IL6, only control mice displayed detectable IL6-immunopositive cells, mainly localised in the PL of DG and also in the CA4 and CA1 subfields of the CA area (Figure 6 and Appendix A). 

The quantitative examination revealed a significant lowering of the density of glial cells expressing GFAP and IL6 in all examined hippocampal regions. Specifically, a significant decrease in GFAP-immunopositive cell density was shown in the DG-GL region of He1-treated mice vs. controls (Figure 7, Panel A, *p* < 0.0001). A similar trend was highlighted in all CA subregions of He1-treated animals, in which a significant decrease in GFAP-immunoreactive cell density was determined (Figure 7, Panel A, *p* < 0.0001). Additionally, an overall significant reduction in IL6-immunopositive cell density was measured in He1-treated mice (Figure 7, Panel C, *p* < 0.0001). To investigate the inflammatory state in aged mice, we also examined the co-expression of immunoreactivity for both GFAP and IL6. Interestingly, a significant decrease of cells showing the double GFAP- and IL6-immunopositivity, namely cell density, was found both in DG and CA subfields of He1-treated mice compared to control animals (Figure 7, Panel E, *p* < 0.001).

Considering every single immunoreactivity, a significant decrease in GFAP and IL6-immunopositive cell OD was measured in the DG of He1-supplemented mice compared to controls (Figure 7, Panel B and D, GL: *p* < 0.0001 and PL: *p* < 0.0001). Concerning the same assessment in the CA area, namely GFAP and IL6-immunoreactive cells, OD evaluation, a region-specific expression, was revealed in He1-treated animals compared to controls. GFAP expression was significantly decreased in all CA regions, with the most marked effect observed in the CA4 and CA2 subfields of He1-treated animals (Figure 7, Panel B, CA4: *p* < 0.0001; CA3: *p* < 0.01; CA2: *p* < 0.0001; CA1: *p* < 0.05). Conversely, the evaluation of IL6-immunopositive cell OD disclosed an extremely significant lessening in CA4 and CA1 regions only (Figure 8, panel D, *p* < 0.0001).

#### 3.4.5. He1 Supplement Counteracts Autophagy Pathway Activation

The presence and distribution of p62 have been presently assessed as a crucial protein involved in the gentle tuning among survival and cell death, and it is also crucially involved in some diseases, including age-related neurodegenerative diseases [50]. The p62 expression pattern highly differed when comparing the two experimental groups. In fact, intensely immuno-marked neurons were observed in the PL of DG in control animals only, while a few weakly immunopositive granule cells were evidenced both in controls and He1-treated mice. As regards the CA area evaluations, similar to the above-reported data concerning microglia, cell immunoreactivity varied depending on the considered CA subfield. In particular, in control mice, the pyramidal cells in CA4, CA3, and CA1 regions showed strong immunolabeling, with many cells characterised by an intensely immunopositive large soma. Differently, very weak labelling was detected in the CA2 region pyramidal cells both in control animals and He1-treated mice (Figure 8 and Appendix A). The following quantitative analysis disclosed that p62-immunoreactivity, measured in both cell density and OD, significantly decreased in He1-treated mice compared to control mice in GL and PL of DG (Figure 8, Panels A and B). Showing an analogous tendency, even with more pronounced effects, the p62-immunoreactivity, considering both cell density and OD, significantly diminished in the CA area of He1-supplemented mice vs. controls (Figure 8, Panels A and B).

#### 3.4.6. He1 Supplement Reduces Cellular Senescence

Histone 2AX (H2AX) is among the initial molecules engaged in the response to DNA damage. Following exposure to agents that damage DNA, H2AX undergoes fast phosphorylation/activation, resulting in γH2AX, which is crucial for preventing genomic instability [51]. Our results showed the absence of γH2AX expression in the DG and CA areas (both for immunopositive cell density and OD) of control and He1-treated mice (Figure 9). Particularly, on the contrary, observing the cerebral cortex as a reference CNS area known to express γH2AX, an evident immunoreactivity was noticeable, with the following quantitative examination corroborating a significant increase of γH2AX-immunolabeling OD in He1-supplemented mice vs. controls (Figure 9, *p* < 0.0001)

## 4. Discussion

The intestinal microbiota plays a vital role in several physiological processes, such as metabolism, immune system modulation, and neurodevelopment. Recent research has unveiled a relationship between gut microbiota composition and memory formation and its impact on systemic- and neuro- inflammation [52]. The gut–brain axis, the bidirectional communication network between the gut and the CNS, has recently attracted considerable attention. In particular, recent studies have shown that certain microbial species present in the gut can have effects on cognitive function and memory formation [33]. The gut microbiota composition has been associated with alterations in neurotransmitter levels and the production of metabolites, which can affect neuronal plasticity and memory consolidation [53]. Indeed, gut microbiota dysbiosis has been associated with cognitive impairment and increased risk of neurodegenerative diseases [54].

Inflammation within the CNS, defined as neuroinflammation, has been related to various neurodegenerative diseases and cognitive impairment [55]. Emerging evidence suggests that gut microbiota dysbiosis can contribute to neuroinflammation by promoting the translocation of pro-inflammatory molecules and pathogens across, in sequence, the intestinal barrier and the blood–brain barrier, reaching the CNS and triggering a cascade of inflammatory processes [56,57]. Chronic systemic inflammation, often associated with gut microbiota dysbiosis, has been linked to cognitive decline and memory impairment [58]. Moreover, prolonged exposure to inflammation can lead to reduced hippocampal neurogenesis, a fundamental process for the consolidation of new amnestic information [59].

The modulation of gut microbiota composition through various interventions has shown promise in improving memory function and reducing inflammation. Probiotics, prebiotics, and dietary changes have been explored as approaches to restore a eubiotic gut microbial community. In particular, prebiotics encourage the growth of advantageous gut bacteria and indirectly affect cognition and inflammation [60]. Medicinal mushrooms can be repurposed as a potential and sustainable source of prebiotics thanks to their high content of bioactive polysaccharides [61]. 

In our previous works, we have demonstrated that an *H. erinaceus* extract (He1) boosted partial recovery on both locomotor and cognitive decline during physiological ageing in frailty mice [38,39]. In the present study, we focused on the association among gut microbiota composition, hippocampal cognitive performances and structural morphology, inflammaging and senescence markers in the hippocampus.

Firstly, we confirmed that He1 oral supplementation during senescence can partially revert cognitive frailty in mice, improving recognition memory performances and protecting cognitive decline during physiological ageing. Secondly, we investigated the relationship between cognitive frailty and the gut microbiota composition in aged mice. In detail, the frail aged mice displayed a lower alpha-diversity index compared to no-frail aged control animals. Next, we demonstrated that *H. erinaceus* two-month oral supplementation in frailty mice did not change the overall gut microbiota composition in a significant manner, as demonstrated by a similar alpha- and beta-diversity in control and He1 groups (T1 vs. T2, pre and post-treatment). Consistent with this result, several literature studies reported that probiotic or prebiotic oral supplementation in the elderly did not modify overall gut microbiome composition [62]. 

However, *H. erinaceus* oral supplementation significantly changed the relative abundance of few but important genera, with a significant reduction of *Odoribacter*, *Clostridia vadinBB60*, and *Muribaculaceae* and a significant increase in the relative abundance of *Clostridia UCG-014*, *Lachnospiraceae_NK4A136*, and *Eubacterium xylanophilum* compared to untreated mice. Table 2 and Table 3 listed previously reported effects on host health for each bacterium, focusing on their relationship with inflammation and cognitive performances. Like many others, the role of these bacteria as dysbiotic or eubiotic genera is controversial. It can be hypothesised that the multiple species of the reported genera play different roles in the gut, which, nonetheless, still needs more exploration. However, all together, comparing and integrating our results with literature papers, we can speculate that *Odoribacter*, *Clostridia vadin_BB60*, and *Muribaculaceae* are negatively correlated with healthy cognitive performance, also increasing the neuroinflammation, whereas *Clostridia UCG-014*, *Lachnospiraceae NK4A136*, and *Eubacterium xylanophilum* could contribute to decreasing neuroinflammaging, determining better cognitive performances in elderly mice.

It should be noted that the genera that changed after He1 supplementation are different from those that we previously described in a recent paper, a longitudinal study of microbiota composition changes during ageing, with particular focus on cognitive frailty development [29]. Specifically, in that mentioned paper, we suggested that some genera (eubiotic bacteria) changed in association with the increased cognitive frailty, attempting to counterbalance the cognitive frailty development, whereas other genera are probably dysbiotic in function. He1 supplementation selectively enhances and, alternatively, decreases different genera, showing a selective plasticity of microbiota composition and those changes are associated with an improvement in memory recognition performance.

Altogether, the gut microbiome results highlighted a possible anti-inflammatory effect of He1 supplementation in elderly frail mice. 

Based on these speculations and considering the known bidirectional network between gut microbiota and CNS, we investigated different neuroinflammation hallmarks in the hippocampus, one of the more sensitive brain areas to neurodegeneration and neuroinflammaging [46]. Previous research showed that structural/functional alterations and changes in the hippocampus have been related to the development and/or the severity of neurodegenerative disorders that cause cognitive decline. 

All immunohistological evaluations were performed at T2, comparing elderly He1-treated and control mice. It should be mentioned, once more, that at T1, we decided to treat with He1 more frail mice, as defined by our cognitive frailty index and at T2, due to the He1 supplementation, the cognitive frailty index of the two experimental groups was not statistically different.

Histological evaluations with H&E evidently demonstrated an age-related alteration in control mice during physiological ageing, in which both the DG and CA subfields displayed a greater number of shrunken granular and pyramidal neurons. Notably, a decrease in altered hippocampal neurons was measured in He1-treated mice, paralleled by a significant decrease in fibrotic response, thus suggesting a significant protective role of He1 supplementation in frail mice. Shrunken neurons in the DG-GL could potentially affect neurogenesis, an event occurring in the subgranular zone (SGZ) of the hippocampal DG during ageing [114]. Several studies demonstrated that age negatively correlates with hippocampal neurogenesis, leading to a progressive impairment of memory formation and consolidation [10]. Our data agree with previous experimental evidence demonstrating that He1 supplementation promotes the restoration of cell proliferation in DG granule cells and adult hippocampal neurogenesis in the adult mouse brain [38,115]. Several studies showed structural changes in CA1 and CA3 subregions of rodents during ageing [116]. Additionally, in our previous work, we defined the effect of He1 supplementation in adult mice on increasing recognition memory performances and ameliorating the glutamatergic neurotransmission in the DG-CA3 synapse recorded in patch–clamp configuration [40]. According to literature data, ageing mouse models exhibit compromised extracellular matrix (ECM) and vascular organisation, which can impact their overall health [117]. Collagens, a key element of the ECM, have a crucial role in this process. Furthermore, neurovascular changes in the hippocampus may contribute to decreased neurogenic capacity and increased neuronal loss, which are commonly observed with age. In the current paper, PSR results demonstrated a reduction in collagen expression following the He1 supplementation in both DG and CA subregions near the endothelium of parenchymal blood vessels. This alteration in ECM protein organisation could impair the signalling pathways needed for synaptic transmission, potentially contributing to cognitive decline in ageing. Additionally, older mice have been found to experience a worsening of the neurovascular structure, with a decrease in the basement membrane and the loss of pericytes that can weaken the blood–brain barrier (BBB) and lead to vascular leakage [118]. When the BBB becomes compromised, harmful neurotoxic proteins can enter the CNS, leading to neuroinflammation, oxidative stress, and cell death, all phenomena linked to early neurodegeneration and cognitive decline [119].

A bulk of the literature has well established that neuroinflammation is a common feature and a potential mediator of cognitive impairments in the elderly. Ageing influences glial reactivity, contributing to the onset of a chronic neuroinflammation scenario called “inflammaging” that contributes to the worsening of the CNS microenvironment, thus accelerating cognitive impairment [120]. Microglial cells are abundant in the hippocampus and have a crucial role in the innate immune system; however, they may trigger extreme or prolonged neuroinflammation, which leads to a morphological and functional shift from resting microglia to the activated and hypersensitive phenotype [121,122]. CD45 is the most abundant protein tyrosine phosphatase expressed on all immune cells, which has a critical role in immune response regulation. Numerous studies demonstrated that CD45 expression increases in microglia with inflammation and ageing [48,123]. According to the literature, our current results demonstrated the protective role of the He1 two-month oral supplementation, activating a partial microglial activation recovery in elderly mice. Specifically, we proved a significant CD45 reduction in supplemented animals in all the hippocampal areas considered. These results linked to microglial activation were additionally corroborated by data on GFAP and IL6, recognised to be elective markers of reactive gliosis and neuroinflammation. In the elderly, the immune system undergoes remodelling due to cytokine dysregulation, which can lead to inflammaging. Our current findings show a decreased expression of both GFAP and IL6 in the DG and CA subfields of the hippocampus in supplemented mice, demonstrating a reduction in inflammation. These results suggest that age-related reactive gliosis may be due to an elevated number and/or activity of astrocytes [124]. Overall, our data show that He1 supplementation has influential effects on aged mice, decreasing neuroinflammation, ameliorating cognitive impairment, and improving elderly quality of life [38].

In ageing brains, there is a noticeable increase in polyubiquitinated proteins within neurons, particularly in neurodegenerative diseases [125]. Autophagy is a degradation pathway for organelles and aggregated proteins that, in the brain, are necessary for memory formation [126]. p62 is a protein strictly involved in the autophagy process and protein turnover. Autophagy and p62 are mutually dependent components of the protein regulation system, working together to maintain cellular homeostasis [50,127]. The role of p62 as an adaptor for protein signalling and its control over the autophagic degradation of certain binding partners is becoming increasingly evident. Loss of p62 results in accelerated ageing due to impaired proteostasis, dysregulated signalling pathways, and inadequate response to oxidative stress. Conversely, p62 overexpression and extensive formation of autophagosomes/autophagic vacuoles are detrimental to neuronal homeostasis. The levels of p62 protein generally rise during ageing, indicating a gradual decline in the autophagic activity of cells [128]. Abnormal metabolism in the hippocampus has been previously related to the progression of neurodegenerative disorders, i.e., Alzheimer’s disease [129].

In this study, we demonstrated that the He1 supplement was strikingly effective in decreasing p62 expression levels (both regarding cell density and OD) in hippocampal DG and CA subfields. Taken together, these results highlight that the reduction in autophagy activation may be implicated in the recovery of cytoarchitectural features observed in the current study. In line with this hypothesis, some evidence suggested that a decrease in the autophagic pathway results in less dendritic pruning and increased dendritic growth in a mouse ageing model [130].

Furthermore, we decided to evaluate the expression of γH2AX, the phosphorylated form of histone H2AX, based on its crucial involvement in DNA damage response and neurodegeneration. Recently, the role of this marker in cellular senescence and ageing has become increasingly evident. Intriguingly, the expression pattern of γH2AX varies depending on the brain region considered. This protein is highly expressed in layer II of the cerebral cortex and in the neurogenic niche of the brain. A lesser extent interests the SGZ of the hippocampal DG [131]. In line with the literature, our data showed the absence of γH2AX expression in the DG and CA subfields (evaluated both for cell density and OD) of control and He1-supplemented mice. On the contrary, in the cerebral cortex, we obtained interesting results indicating that He1-supplementation reduces γH2AX expression, thus potentially modulating DNA damaging and neurodegenerative events occurring during ageing.

The finding that *H. erinaceus* positively modulated gut microbiota composition well corroborated the decreased inflammation. Indeed, accumulating evidence suggests the fundamental role of gut microbiota-derived molecules in the pathogenesis of neurodegenerative diseases. Microbiota’s gut metabolites could reach brain tissue through the gut–brain axis and cross a “leaky” BBB, typically observed in elderly people. This event could consequently trigger the release of toxins and pathogen components that ultimately could affect CNS homeostasis [57].

## 5. Conclusions

The community of microorganisms living in the gut, known as the gut microbiome, is intricate and ever-changing. Research on its impact on various health facets, including cognition and memory, is an active and evolving field. Altogether, our present results pave the way to a prebiotic role of *H. erinaceus*, which is effective in modulating the gut microbiota’s intestinal homeostasis in a complex way, also suggesting a psychobiotic effect. This effect was paralleled by an amelioration in the cognitive frailty index and a general decrease in inflammaging in the hippocampus, which is crucial for memory creation and consolidation. In conclusion, the capabilities of *H. erinaceus* in regulating microbiota composition, improving cognitive performances, and reducing neuroinflammation give strength to the theories, which are based on microbiome modulation therapies with age-delaying and longevity-promoting effects.

## Figures and Tables

**Figure 1 biology-13-00018-f001:**
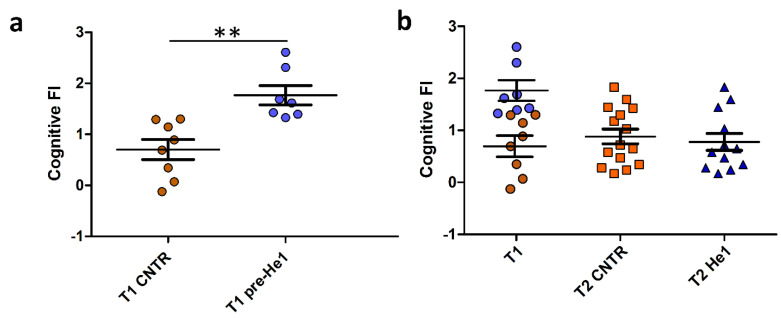
He1 treatment reverted cognitive frailty evaluated as the knowledge component of recognition memory during physiological ageing in mice. (**a**): Scatter plot of Cognitive FI Values measured at T1 in control (T1 CNTR, orange) and in pre-supplemented He1 mice (T1 pre-He1, blue). (**b**): Scatter plot of Cognitive FI Values measured at T1 in control and in pre-supplemented He1 animals (T1, orange and blue, respectively), and at T2 in control (T2 CNTR, orange) and in He1-supplemented mice (T2 He1, blue). *P*-value was calculated by unpaired Student’s *t*-test: *p* < 0.01 (**).

**Figure 2 biology-13-00018-f002:**
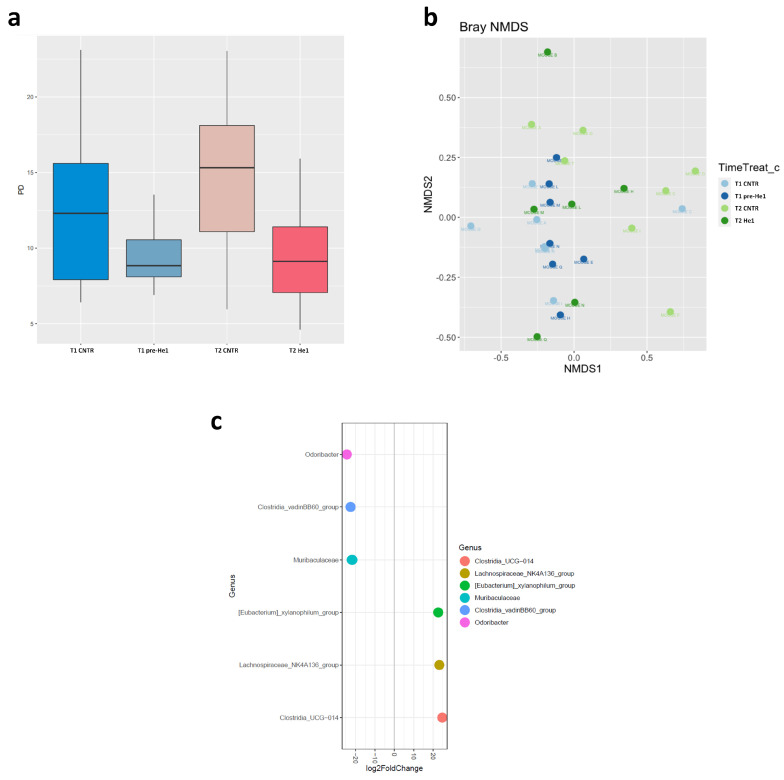
Gut microbiome composition. (**a**): Alpha-diversity distribution box plots estimated as Faith’s phylogenetic distance (PD) in control mice and He1-treated animals at T1 and T2 (T1 CNTR: blue; T1 pre-He1: light blue; T2 CNTR: pink; T2 He1: fuchsia). (**b**): Non-metric multidimensional scaling (NMDS). Colours in the bidimensional NMDS plot are used, as shown in the legend. The ordinate analysis is founded on the Bray–Curtis distance matrix. The graphical plot and the ellipses were generated by the ggplot2 R package implemented with the stat ellipse function. (**c**): Differential abundance (of selected genera that significantly changed) at T2 in He1 treated vs. control mice.

**Figure 3 biology-13-00018-f003:**
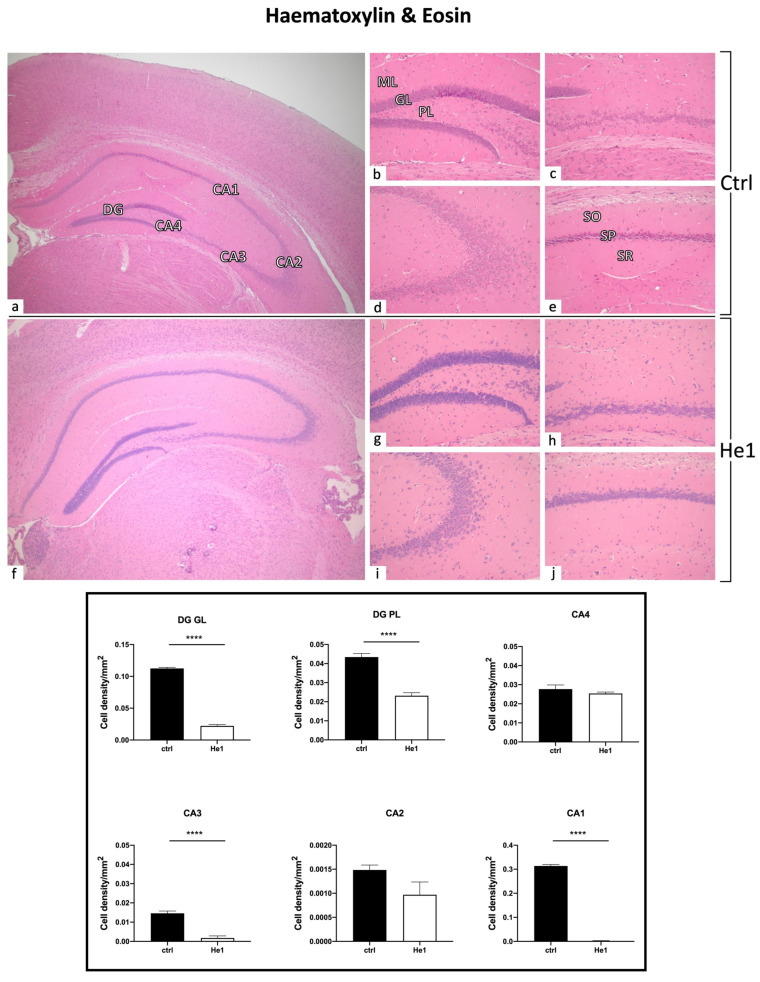
H&E staining revealing the well-preserved physiological hippocampal cytoarchitecture in non-supplemented controls (**a**–**e**) and He1-treated (**f**–**j**) elderly mice. (**a**,**f**): Low magnification micrograph shows the whole hippocampus, formed of *Cornus Ammonis* (CA; subdivided into CA1, CA2, CA3, and CA4) and dentate gyrus (DG). (**b**,**g**): Higher magnifications of the DG area displaying three distinct layers: ML, GL, and PL. (**e**,**j**): Higher magnifications of the CA1 region, showing the typical three-layered structure: outer polymorphic layer, i.e., *Stratum oriens* (SO); middle pyramidal cell layer, i.e., *Stratum pyramidale* (SP); inner molecular layer, i.e., *Stratum radiatum* (SR) in control and He1-supplemented animal, respectively. Light microscopy magnification: 4× (**a**,**f**); 20× (**b**–**e**,**g**–**j**). Histograms display the quantitative valuation of shrunken cell density in DG and CA subregions. *p*-values calculated by unpaired Student’s *t*-test. **** *p* < 0.0001.

**Figure 4 biology-13-00018-f004:**
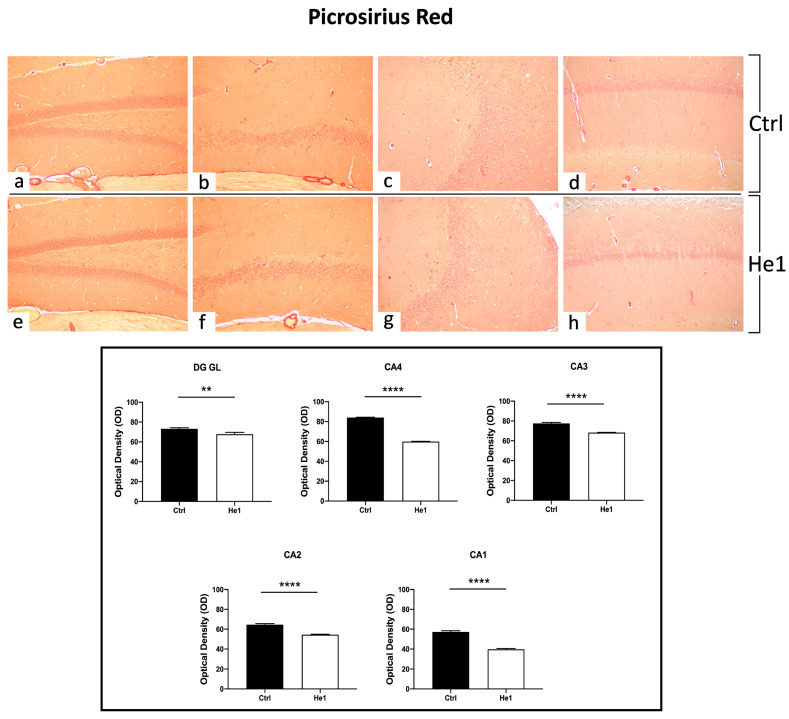
PSR staining evaluation under light microscopy. Representative hippocampal specimens showing parenchyma and blood vessels from control (**a**–**d**) and He1-treated animals (**e**–**h**). Microscopy magnification: 20× (**a**–**h**). Panel A: Histograms showing OD measurements in DG and CA subregions. *p*-values calculated by unpaired Student’s *t*-test. *p*-values: ** *p* < 0.01; **** *p* < 0.0001.

**Figure 5 biology-13-00018-f005:**
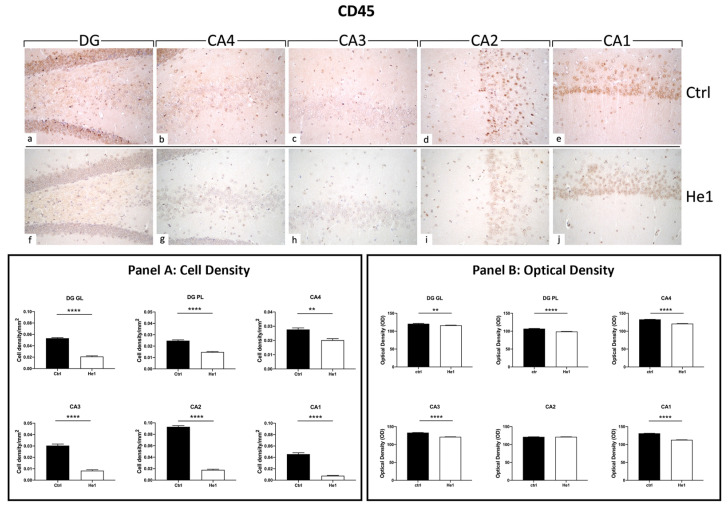
DAB-immunostaining reaction for CD45 in hippocampal DG and CA subfields from control mice (**a**–**e**) and He1-treated mice (**f**–**j**). Magnification: 40× (**a**–**j**). Panel A: Histograms depicting CD45-immunopositive cell density assessed in hippocampal DG and CA subregions of control and He1-supplemented mice. *p*-values calculated by unpaired Student’s *t*-test. ** *p* < 0.01; **** *p* < 0.0001. Panel B: Histograms display CD45-immunopositive cell OD measured in hippocampal DG and CA subregions of control and He1-supplemented mice. *p*-values calculated by unpaired Student’s *t*-test. ** *p* < 0.01; **** *p* < 0.0001.

**Figure 6 biology-13-00018-f006:**
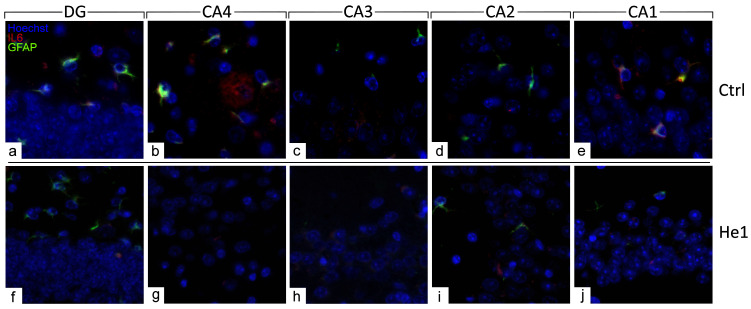
Double immunohistochemical detection of GFAP (green signal) and IL6 (red signal) by fluorescence microscopy in control animals (**a**–**e**) and He1-treated (**f**–**j**) mice. DNA counterstaining with Hoechst 33258 (blue fluorescence). Magnification: 60× (**a**–**j**).

**Figure 7 biology-13-00018-f007:**
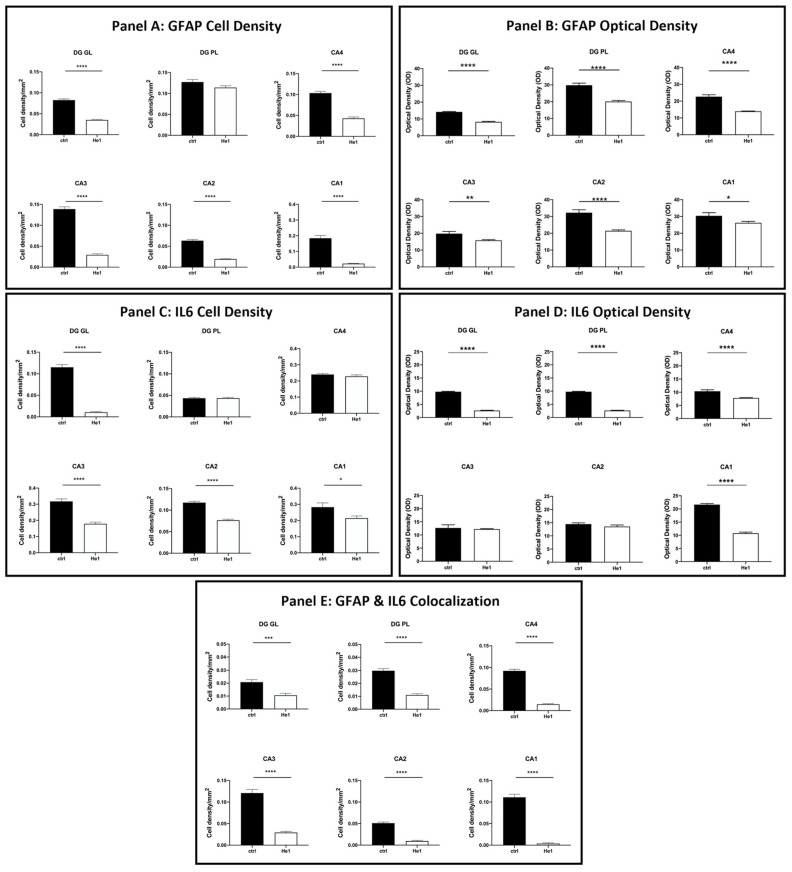
Panel A: Histograms illustrating GFAP-immunopositive cell density determined in DG and CA subregions of control and He1-supplemented mice. *p*-values determined by unpaired Student’s *t*-test. **** *p* < 0.0001. Panel B: Histograms indicating GFAP-immunopositive cell OD measured in DG and CA subregions of control and He1-supplemented mice. *p*-values calculated by unpaired Student’s *t*-test. * *p* < 0.05; *** p*<0.01; **** *p* < 0.0001. Panel C: Histograms illustrating IL6-immunopositive cell density determined in DG and CA subregions of control and He1-supplemented mice. *p*-values calculated by unpaired Student’s *t*-test. * *p* < 0.05; **** *p* < 0.0001. Panel D: Histograms illustrating IL6-immunopositive cell OD determined in DG and CA subregions of control and He1-supplemented mice. *p*-values calculated by unpaired Student’s *t*-test. **** *p* < 0.0001. Panel E: Histograms showing the GFAP- and IL6- double-immunopositive cell density, determined in DG and CA subregions of control and He1-supplemented mice. *p*-values calculated by unpaired Student’s *t*-test. *** *p* < 0.001; **** *p* < 0.0001.

**Figure 8 biology-13-00018-f008:**
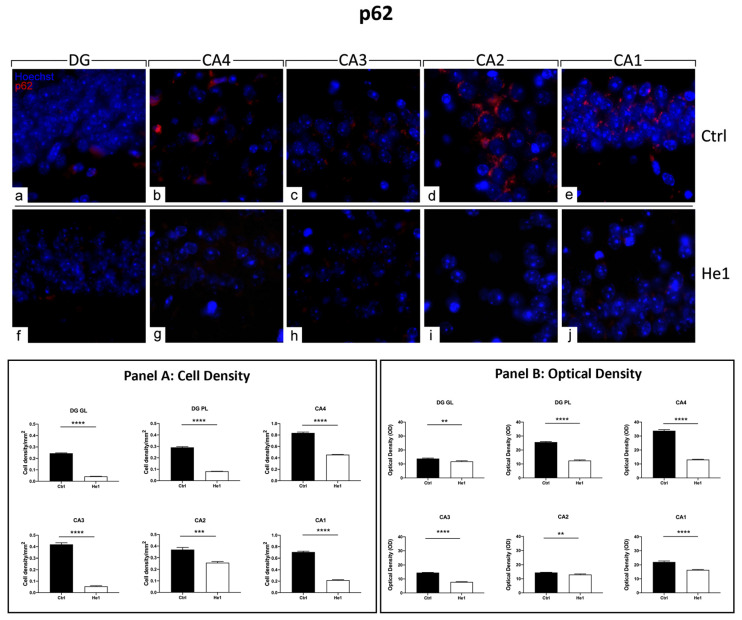
Immunocytochemical detection of p62 (red signal) by fluorescence microscopy in control animals (**a**–**e**) and He1-treated (**f**–**j**) mice. DNA counterstaining with Hoechst 33258 (blue fluorescence). Magnification: 60× (**a**–**j**). Panel A: Histograms displaying p62-immunopositive cell density measured in DG and CA subregions of control and He1-supplemented mice. *p*-values calculated by unpaired Student’s *t*-test. *** *p* < 0.001; **** *p* < 0.0001. Panel B: Histograms presenting p62-immunopositive cell OD determined in DG and CA subregions of control and He1-supplemented mice. *p*-values calculated by unpaired Student’s *t*-test. ** *p* < 0.01; **** *p* < 0.0001.

**Figure 9 biology-13-00018-f009:**
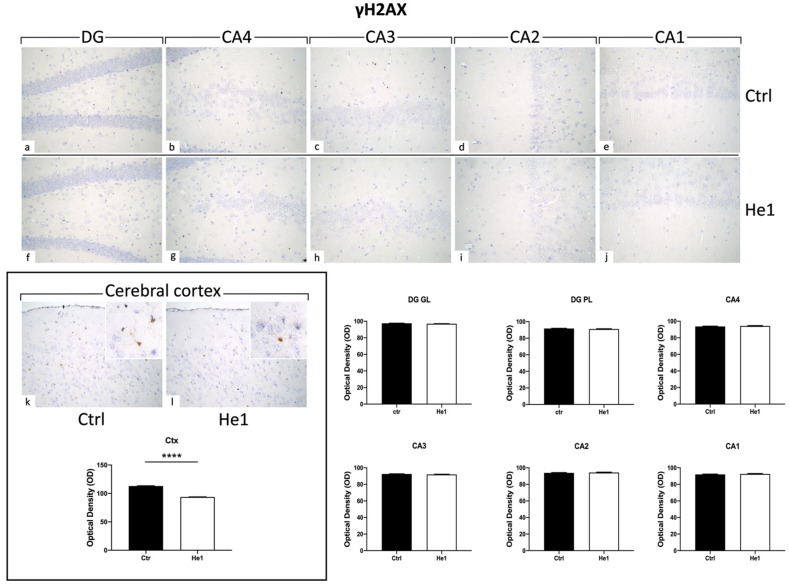
Representative micrographs depicting immunohistochemical reaction for γH2AX in hippocampal DG and CA subfields and in cerebral cortex from controls (**a**–**e**,**k**) and He1-treated mice (**f**–**j**,**l**). Magnification: 40× (**a**–**l**); 100× (insert in **k**,**l**). Histograms illustrating γH2AX-immunopositive cell OD assessed in hippocampal DG and CA subregions and in cerebral cortex of control and He1-supplemented mice. *p*-values calculated by unpaired Student’s *t*-test. **** *p* < 0.0001.

**Table 1 biology-13-00018-t001:** Amounts of bioactive metabolites present in 1 g (dried weight) of He1 mycelium and sporophore.

Bioactive Metabolites	He1	
	Sporophore	Mycelium
Erinacine A (µg/g)	-	150
Hericenone C (µg/g)	500	-
Hericenone D (µg/g)	<20	-
L-Ergothioneine (µg/g)	340	580

**Table 2 biology-13-00018-t002:** Published effects on host health of gut genera that significantly decreased with two months of He1 oral supplementation in frail mice. In particular, we examined the literature for the involvement of these genera on host health, focusing on CNS and inflammation.

**Bacterium** **Genera**	**Effects on Host Health**	**Ref.**
*Odoribacter*	Dichotomous role:	
	SCFAs producer.	[63,64,65,66]
	Opportunistic pathogen with a potential correlation with inflammation, increasing the levels of pro-inflammatory cytokines.	[67,68,69]
	Increased in (i) people with Alzheimer’s Disease, (ii) Parkinson’s disease patients with cognitive impairment, (iii) major depression patients with cognitive impairment, (iv) patients with post-finasteride treatment-associated cognitive and psychological disorders, (v) women with HIV and cognitive impairment, and (vi) during physiological ageing in wild-type mice with an impairment in spatial memory and anxiety-like behaviours.	[58,70,71,72,73,74]
	*Odoribacter* exhibits a protective association with cognitive impairment and hippocampal volume.	[75,76]
*Clostridia vadinBB60*	Dichotomous role:	
	Commensal beneficial bacteria.	[77,78]
	Its increase is associated with neuroplasticity decrease in schizophrenic patients and elderly frail mice.	[29,79]
*Muribaculaceae*	Dichotomous role:	
	SCFAs producer.	[80,81]
	Its relative abundance is positively correlated with IL-1β in sleep-deprived mice. It could contribute to LPS production, and a reduction in *Muribaculaceae* relative abundance determined an anti-inflammatory effect, decreasing the expression of TNF-α and IL-6 in diabetic mice.	[82,83]
	It might disrupt the microglial M1/M2 phenotype ratio homeostasis after chronic methamphetamine exposure in mice.	[84]
	It could lead to learning and memory deficits after prolonged methamphetamine usage in mice.	[84]

**Table 3 biology-13-00018-t003:** Published effects on host health of gut genera that significantly increased with two months of He1 oral supplementation in frail mice. In particular, we examined the literature for the involvement of these genera on host health, focusing on CNS and inflammation.

Bacterium Genera	Effects on Host Health	Ref.
*Clostridia UCG-014*	Dichotomous role:	
	SCFA producer, reducing gut and neuro-inflammation, communicating with the immune system, and strengthening the gut barrier in different preclinical and clinical models.	[85,86,87,88,89,90]
	Pro-inflammatory bacterium.	[91,92,93,94]
*Lachnospiraceae_NK4A136*	SCFA (butyrate) producer and possible role in bile acids metabolism and cholesterol homeostasis. Association among higher abundance of *Lachnospiraceae NK4A136*, enhancement of gut barrier function, and anti-inflammatory properties.	[95,96,97,98,99,100,101,102,103]
	Reduced in patients with dementia or cognitive impairments.Its increase is correlated with improved memory performance.	[97,104,105]
	It is involved in the production of a few neurotransmitters, which play crucial roles in memory and cognitive function.	[106,107]
*Eubacterium xylanophilum*	SCFA (formic, acetic, and butyric acids) producer.	[108,109,110,111]
	An association between memory indicators and gut microbiota metabolites produced by *Eubacterium xylanophilum* was demonstrated in preclinical models.	[112,113]

## Data Availability

The sequences that were generated in this study have been deposited into the EMBL-EBI database in project PRJEB54046.

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
