# Peer review of "Hericium erinaceus Extract Exerts Beneficial Effects on Gut–Neuroinflammaging–Cognitive Axis in Elderly Mice"

_biology, 2023, doi:10.3390/biology13010018_

Round 1

Reviewer 1 Report

Comments and Suggestions for Authors

The manuscript is carefully written and describes valuable results. The following points should be corrected:

-The title should be changed. It should be clear from the title that the results were obtained on mice.

-line 152: 1 mg supplement/mouse daily: Dry weight ?

-line 304: 580 µg/g L-ergothionein

Author Response

Responses to Reviewer #1’s comments

The manuscript is carefully written and describes valuable results. The following points should be corrected:

Q1: The title should be changed. It should be clear from the title that the results were obtained on mice.

AA: We thank the reviewer for his/her suggestion. We accordingly modified the title: “Hericium erinaceus extract exerts beneficial effects on gut-neuroinflammaging-cognitive axis in elderly mice”.

Q2: line 152: 1 mg supplement/mouse daily: Dry weight?

AA: We thank the reviewer for underlying this lack. We added this specific information.

Q3: line 304: 580 µg/g L-ergothioneine

AA: We thank the reviewer for pointing out this mistake. We fittingly modified text.

We would like to thank the Reviewer#1 for his/her time spent on reviewing our manuscript and his/her valuable comments, helping us improving the article. We have carefully revised the manuscript following his/her suggestions. We hope that the made changes addressed all the points and concerns raised by the Reviewer. All made changes or improvements are highlighted in YELLOW color throughout the text. Thanks again.

Reviewer 2 Report

Comments and Suggestions for Authors

Authors reported the complex prebiotic role of  H. erinaceus also suggesting a psychobiotic role. 

Given the complexity of the action of the H. erinaceous on the composition of the gut microbiome in mice, how do the Authors think to transpose these results into clinical setting ( real world)?

Will it be possible on the elderly with cognitive impairment to assess the real action of H.erinaceous on their microbiome ?

Author Response

Responses to Reviewer #2’s comments

Authors reported the complex prebiotic role of H. erinaceus also suggesting a psychobiotic role.

Q1: Given the complexity of the action of the H. erinaceus on the composition of the gut microbiome in mice, how do the Authors think to transpose these results into clinical setting (real world)?

Q2: Will it be possible on the elderly with cognitive impairment to assess the real action of H. erinaceus on their microbiome?

AA: We thank the Reviewer#2 for rising these interesting issues.

It has to be underlined that, using our preclinical in vivo model, particular attention was devoted to choose a translational approach during the design of the experimental plan and the execution of all experimental procedures. Specifically, in the current work:

  • the He1 dose was properly selected to mimic the human intake;
  • the spontaneous behavioral tests were chosen based on their resemblance to those used in clinical setting. Indeed, the recognition memory in humans is usually assessed using Sternberg Item Recognition Paradigm, a test very similar to NOR and emergence tasks (Brodziak et al., 2014);
  • the gut microbiome composition was investigated employing the same methodology that clinicians can use to study human gut microbiota composition (National Academies of Sciences, Engineering, and Medicine et al., 2017).

Interestingly, assessing the real action of H. erinaceus on the gut microbiome of elderly individuals with cognitive impairment could be challenging but feasible. It has to be reminded that, even though the nootropic and neuroprotective properties of H. erinaceus are well-known, further research is still needed to establish a clear understanding of H. erinaceus effects on the gut microbiome and its relationship with cognition in elderly.

It is has to be taken into careful consideration that elderly population, often characterized by mild to severe cognitive impairment, may have been diagnosed for different pre-existing pathologies/disorders/comorbidities, and therefore may have been in therapy with diverse medications. Both pathologies/disorders/comorbidities as well as the relative conventional pharmacological treatment protocols may affect the composition of gut microbiome, also interacting with a potential H. erinaceus oral supplementation. Therefore, it is important to carefully consider the individual's specific health status together with potential risk factors. In particular, concerning human gut-microbiome-brain axis study, the enrollment of frail elderly patients should be achieved fixing peculiar recruitment and eligibility (inclusion/exclusion) criteria. For example, analyzed cohorts should be characterized/stratified based on age, sex, disorders, comorbidities, pharmacological treatment, cognitive performances (e.g. evaluation using with MMSE test), etc.

To date, in literature few studies exist investigating the effect of H. erinaceus on human cognition. A recent review by Brandalise et al. (2023) summarized the available literature regarding this topic, suggesting that the H. erinaceus supplementation improved human cognition in MCI and early AD diagnosed patients.

Figure from Brandalise et al., 2023.

Moreover, it has to be considered that, due to the plasticity of the gut ecosystem, a longitudinal study evaluating both cognitive performance and microbiota composition (through stool sample analyses) is needed to study gut microbiome-brain axis modulation in a proper manner.

Notably, the clinical investigations reported in the figure are pilot studies conducted on a small number of patients. Since the gut microbiome is largely dependent from epigenetic factors, such as diet and lifestyle, studies on larger sets of subjects should be preferable conducted.

Compared to human studies, it has to be highlighted that the use of our preclinical in vivo model could partially overcame the lifestyle interference which may influence the gut microbiome heterogeneity, allowing us to precisely discriminate the H. erinaceus effects.

In addition, few literature data reported the effect of H. erinaceus on gut microbiome composition in humans (Xie et al., 2021; Zhuang et al., 2023), focusing on the polysaccharide portion of H. erinaceus. Differently, as reported in our manuscript - Results section, our standardized extract was obtained with an ethanolic-water extraction which allowed us to obtain an extract enriched in nootropic metabolites, rather than polysaccharides. Therefore, before its employment in clinical setting, the selected H. erinaceus supplement will have to be quantified/standardized for all bioactive compounds, such as nootropic and polysaccharide components.

All these considerations lead us to suggest the need for a multidisciplinary approach, making available complementary skills and know-how, therefore involving preclinical researchers, microbiologists, clinicians, and biotechnologists to transpose the results of preclinical research from mouse models into clinical settings.

Our team is always open to collaboration in human clinical investigations, as already done during clinical studies on depressed/obese/overweight patients (see Vigna et al., 2019).

Once again, we would like to thank the Reviewer#2 for his/her time spent on reviewing our manuscript and his/her valuable comments, helping us improving the article. We have carefully revised the manuscript following his/her suggestions. We hope that the made changes addressed all the points and concerns raised by the Reviewer. All made changes or improvements are highlighted in YELLOW colour throughout the text. Thanks again.

Brandalise F, Roda E, Ratto D, Goppa L, Gargano ML, Cirlincione F, Priori EC, Venuti MT, Pastorelli E, Savino E, Rossi P. Hericium erinaceus in Neurodegenerative Diseases: From Bench to Bedside and Beyond, How Far from the Shoreline? J Fungi (Basel). 2023 May 10;9(5):551. doi: 10.3390/jof9050551. PMID: 37233262; PMCID: PMC10218917.

Brodziak A, KoÅ‚at E, Różyk-Myrta A. In search of memory tests equivalent for experiments on animals and humans. Med Sci Monit. 2014 Dec 19;20:2733-9. doi: 10.12659/MSM.891056. PMID: 25524993; PMCID: PMC4280055.

National Academies of Sciences, Engineering, and Medicine; Division on Earth and Life Studies; Board on Life Sciences; Board on Environmental Studies and Toxicology; Committee on Advancing Understanding of the Implications of Environmental-Chemical Interactions with the Human Microbiome. Environmental Chemicals, the Human Microbiome, and Health Risk: A Research Strategy. Chapter 4: Current Methods for Studying the Human Microbiome. Washington (DC): National Academies Press (US); 2017 Dec 29. PMID: 29431953.

Vigna L, Morelli F, Agnelli GM, Napolitano F, Ratto D, Occhinegro A, Di Iorio C, Savino E, Girometta C, Brandalise F, Rossi P. Hericium erinaceus Improves Mood and Sleep Disorders in Patients Affected by Overweight or Obesity: Could Circulating Pro-BDNF and BDNF Be Potential Biomarkers? Evid Based Complement Alternat Med. 2019 Apr 18;2019:7861297. doi: 10.1155/2019/7861297. PMID: 31118969; PMCID: PMC6500611.

Xie XQ, Geng Y, Guan Q, Ren Y, Guo L, Lv Q, Lu ZM, Shi JS, Xu ZH. Influence of Short-Term Consumption of Hericium erinaceus on Serum Biochemical Markers and the Changes of the Gut Microbiota: A Pilot Study. Nutrients. 2021 Mar 21;13(3):1008. doi: 10.3390/nu13031008. PMID: 33800983; PMCID: PMC8004025.

Zhuang H, Dong H, Zhang X, Feng T. Antioxidant Activities and Prebiotic Activities of Water-Soluble, Alkali-Soluble Polysaccharides Extracted from the Fruiting Bodies of the Fungus Hericium erinaceus. Polymers (Basel). 2023 Oct 20;15(20):4165. doi: 10.3390/polym15204165. PMID: 37896408; PMCID: PMC10611342.

Reviewer 3 Report

Comments and Suggestions for Authors

In this study, Cecilia et al. investigated the effects of Hericium erinaceus extract on the gut, neuroinflammation, and cognitive performance in elderly mice. After carefully reviewing this paper, I recognized there were a few issues. Here are some comments on this paper:

1.      Line 26, “H. erinaceus” needs to be italicized, line 31 “longevi-ty-promoting” should be “longevity”.

2.      It is suggested that in line 47, the word " thanks to" be replaced by another word.

3.      Line 144 “FI = (Value - Mean Value at T0) / (Standard Deviation at T0) * ± 0.2” does * mean multiply?

4.      Line 169 “16s rRNA sequencing” should be 16S rRNA gene sequencing.

5.      Line 175 “adapters” should be primers.

6.      It is proposed that Tables 1 and 2 are placed in the supplement files.

7.      In section 2.9, the microbiome data processing was repeated with section 2.4.

8.      Figure 1 a, what is “time_treatmen_c”.

9.      Figure 3 CA2, could the y-axis be scale down?

10.   Given that the immunofluorescence figures 6-8 were rather dark and hard to see, could it be placed in the supplementary file?

11.   It is proposed that Tables 4 and 5 be moved, this study was an Article, rather than a Review.

12.   For the article manuscript, 149 references seem excessive. Authors are advised to streamline the number of references appropriately.

Author Response

Responses to Reviewer #3’s comments

In this study, Cecilia et al. investigated the effects of Hericium erinaceus extract on the gut, neuroinflammation, and cognitive performance in elderly mice. After carefully reviewing this paper, I recognized there were a few issues. Here are some comments on this paper:

Q1: Line 26, “H. erinaceus” needs to be italicized, line 31 “longevity-promoting” should be “longevity”.

AA: We thank the reviewer for his/her suggestions. We accordingly modified the text.

Q2: It is suggested that in line 47, the word " thanks to" be replaced by another word.

AA: We thank the reviewer for his/her suggestions. We accordingly modified the text.

Q3: Line 144 “FI = (Value - Mean Value at T0) / (Standard Deviation at T0) * ± 0.2” does * mean multiply?

AA: We thank the reviewer for raising this question. For clearness, we modified the text replacing the “*” with “×”.

Q4: Line 169 “16s rRNA sequencing” should be 16S rRNA gene sequencing.

AA: We thank the reviewer for his/her suggestions. We accordingly modified the text.

Q5: Line 175 “adapters” should be primers.

AA: We thank the reviewer for his/her suggestions. We accordingly modified the text.

Q6: It is proposed that Tables 1 and 2 are placed in the supplement files.

AA: We thank the reviewer for his/her suggestion. Accordingly, we moved Tables 1 and 2 to supplementary materials.

Q7: In section 2.9, the microbiome data processing was repeated with section 2.4.

AA: We thank the reviewer for his/her comment. Hence, we revised section 2.4 to avoid any repetitions throughout the text.

Q8: Figure 1 a, what is “time_treatmen_c”.

AA: We thank the reviewer for his/her comment. Accordingly, we revised Figure 2a, removing the misprint.

Q9: Figure 3 CA2, could the y-axis be scale down?

AA: We thank the reviewer. Following his/her suggestion, we resized the y-axis to better show the histograms.

Q10: Given that the immunofluorescence figures 6-8 were rather dark and hard to see, could it be placed in the supplementary file?

AA: We thank the reviewer for raising this concern. Nonetheless, it has to be taken into careful consideration that low-magnification micrographs were specifically chosen to allow a simple and clear identification of the different hippocampal areas under investigation, together with a rapid detection of the immunopositive cells. Further, the selected low magnification was chosen to guide the reader in the manuscript comprehension, facilitating a rapid and easy identification of the peculiar topological structures of the hippocampus.

Unfortunately, low magnification images could misleadingly appear as dark, and the immunofluorescence could be apparently scarce. Nonetheless, a strong, specific immunolabeling was clearly observed after all performed reactions, allowing a sharp identification of immunopositive cells, avoiding any interfering noise signal. As result, single immunopositive cells were clearly detectable in specific hippocampal region under investigation.

Therefore, since we are convinced that the current immunofluorescence findings are essential to demonstrate H. erinaceus anti-inflammatory effect in hippocampus of supplemented mice, also following Reviewer’s suggestion, we replaced the previous figures with new high-magnification images for each hippocampal area. Parallelly, we moved the low-magnification figures to supplementary materials, as suggested by the reviewer, with the aim at guiding the reader toward the identification of different hippocampal structures, allowing a better interpretation of the presented anti-inflammatory effect of H. erinaceus.

Q11: It is proposed that Tables 4 and 5 be moved, this study was an Article, rather than a Review.

AA: We thank the reviewer for his/her suggestion. Nonetheless, it has to be highlighted that the two mentioned tables summarized the hypothesized He1 effects on gut microbiome by comparison with the available scientific literature data, focusing on CNS and inflammation. Therefore, we decided to keep the tables in the original position in the text – Discussion section.

Q12: For the article manuscript, 149 references seem excessive. Authors are advised to streamline the number of references appropriately.

AA: We thank the reviewer for his/her comment. With great effort, to meet his/her request, we removed 18 references, trying to maintain the accuracy of pertinent available literature.

We would like to thank the Reviewer#3 for his/her time spent on reviewing our manuscript and his/her valuable comments, helping us improving the article. We have carefully revised the manuscript following his/her suggestions. We hope that the made changes addressed all the points and concerns raised by the Reviewer. All made changes or improvements are highlighted in YELLOW color throughout the text. Thanks again.

Round 2

Reviewer 3 Report

Comments and Suggestions for Authors

The authors fulfilled all requests.